# Factors Affecting the Situational Awareness of Armored Vehicle Occupants

**DOI:** 10.3390/s24113688

**Published:** 2024-06-06

**Authors:** Zihan Pei, Wenyu Zhao, Long Hu, Ziye Zhang, Yang Luo, Yixiang Wu, Xiaoping Jin

**Affiliations:** School of Engineering, China Agricultural University, Beijing 100091, China; 1301631266@cau.edu.cn (Z.P.); 2021307150514@cau.edu.cn (W.Z.); zhangziye2020@outlook.com (Z.Z.); 2021307150509@cau.edu.cn (Y.L.);

**Keywords:** situational awareness, attentional resource allocation, armored vehicles, eye movements, ergonomics

## Abstract

In the field of armored vehicles, up to 70% of accidents are associated with low levels of situational awareness among the occupants, highlighting the importance of situational awareness in improving task performance. In this study, we explored the mechanisms influencing situational awareness by simulating an armored vehicle driving platform with 14 levels of experimentation in terms of five factors: experience, expectations, attention, the cueing channel, and automation. The experimental data included SART and SAGAT questionnaire scores, eye movement indicators, and electrocardiographic and electrodermal signals. Data processing and analysis revealed the following conclusions: (1) Experienced operators have higher levels of situational awareness. (2) Operators with certain expectations have lower levels of situational awareness. (3) Situational awareness levels are negatively correlated with information importance affiliations and the frequency of anomalous information in non-primary tasks. (4) Dual-channel cues lead to higher levels of situational awareness than single-channel cues. (5) Operators’ situational awareness is lower at high automation levels.

## 1. Introduction

Situational awareness (SA) is a term initially used in aviation to mean that pilots are subjected to different environments and respond differently and make different decisions in response to them. Aviation accident studies report that 35.1% of non-major accidents and 51.6% of major accidents are due to pilots’ decision-making failures, which are mainly due to incorrect or missing situational awareness [1]. In the field of armored vehicles, on the other hand, up to 70% of accidents are due to human factors [2]. As emerging technologies continue to evolve, the task itself is becoming less physically taxing on the occupant and is beginning to translate into a drain on cognitive resources [3,4,5], suggesting that maintaining a good level of situational awareness has a critical role to play in improving the task results.

The term situational awareness was first coined by Endsley, who argued that situational awareness is a person’s perception of different elements in a given temporal and spatial environment, their understanding of their meanings, and their projection of their later states. Situational awareness is an individual’s internal representation of the changing external environment, and it is a key factor influencing operator decision-making and performance in complex, dynamically changing information environments [6]. Situational awareness has been categorized into three levels: the first level is for perceiving the current flight environment, the second level is for understanding the current flight situation, and the third level is for making predictions about the flight situation and flight decisions. Several influences that work together on situational awareness have also been proposed, such as attention allocation, long-term working memory, mental modeling, goal and situational awareness, time perception and dynamic changes in each element, and automation levels [7,8,9,10]. In total, situational awareness measures can be categorized into physiological measures, memory probe measures, operational performance measures, and subjective ratings [11]. Some of the more commonly used methods are eye-tracker measurements, comprehensive assessment techniques (freeze measure SAGAT), and self-assessment techniques (SART) [12,13].

Currently, situational awareness is mainly researched in the field of aerospace and autopiloting, and Chuang L. L. and Rasmussen H. B. discussed the concept of situational awareness in the field of navigation and aviation and called for aviation and navigation personnel and leaders to pay attention to the cultivation of situational awareness and decision-making abilities [14,15,16]. Wang A. combined the human information processing process to establish a new situational awareness model and situational circle model to propose the idea of designing a locomotive cab human–machine interface to realize situational awareness [17]. SH Choi investigated and studied the change in situational awareness in helicopter pilots to analyze the identification of factors and countermeasures in emergencies [18]. Durso et al. simulated air traffic control by applying three situational awareness measures, which are the subjective rating (SART), freeze measurements (SAGAT), and real-time measurements, for performance evaluations and comparisons [19]. Zhang T.’s work is based on the combination of multiple resource loading theory and information cognitive processing theory of the Attention Situation Identification (A-SA) model and Attention Resource Allocation SA model, and they propose a new quantitative model of SA, which combines the subjective Ten-Dimensional Situational Awareness Measurement Technique (10-D SART) and the objective Flight Performance, Situational Awareness Comprehensive Measurement Technique (SAGAT), as well as physiological measurements (cardioelectricity, picoelectric and respiratory measurements), for experimental measurements and further application to HUD display interface environments [20,21].

The research field of situational awareness mainly focuses on motor vehicle driving, navigation and aerospace, all of which are based on different driving systems and driving conditions, and different methods are used to measure and analyze the situational awareness of drivers, while there is little research on situational awareness in the field of armored vehicle combat. In situations involving monitoring targets, aiming at targets, and performing loading tasks, artillery firing and other combat tasks, the environment in which special vehicles are located is special, and measuring situational awareness in combination with combat tasks has become a difficult task. Additionally, there are limitations in measuring and warning systems in the narrow spaces of special vehicles. But from a situational awareness measurement aspect, special vehicle combat security is also an urgent problem, and an armored vehicle crew’s situational awareness levels affect the performance of the task. Thus, this study is aimed at the specific implementation of armored vehicle task simulations and the collection of eye movement, skin electricity, heart rate and other physiological signals to explore the impact of the situational awareness on the level of prior knowledge, attention allocation, expectations, and automation level of the armored vehicle crew.

## 2. Method

### 2.1. Subjects

The experiment was conducted with 30 professionals, aged 20–35 (mean age 26, standard deviation plus or minus 2.3), who were male, right-handed, had a normal corrected vision and normal hearing, and were in good physical condition. Normal rest and relaxation before the experiment ensured that they were in a good mental state during the experiment. All subjects were aware of the details of the experiment, and the experiment was approved by the subject’s institution, i.e., the College of Engineering, China Agricultural University, and by the subjects themselves.

### 2.2. Test Equipment

This experimental platform is based on a simulated armored vehicle driving platform, which consists of three parts: the main control system, the auxiliary driving system and the background data recording system. There are two kinds of terrains in the scene, i.e., plains and woods, and there are differences in driving smoothness, enemy obstacle con-cealment and enemy appearance probability. There are two types of enemy obstacles, namely tanker and transportation vehicle. The experimenters included two vehicle cap-tains and drivers. Each experiment lasted about 15 min. The experiment used a Swedish non-contact infrared eye-tracking device, Smart-Eye, to collect eye movement data from the subjects. The device uses external infrared cameras fixed to the sides and top of the display to ensure that subjects can perform the experiment in a completely natural state, as shown in the figure. The experiment was synchronized with the use of an ECG and EKG acquisi-tion system to follow up on the subject’s physiological signals. The test instruments and equipment are shown in Figure 1 (The red line leading from the eye in Figure 1c represents the line of sight, and the remaining red, blue, and green lines are the coordinate axes.).

### 2.3. Test Design

This study used a univariate controlled experimental design aimed at analyzing the effects of different factors on situational awareness. The experimental conditions were set based on two main variables: first, the background differences of the participants, and second, configuration changes of the simulated driving platform. The specific experimental program is shown in Figure 2.

#### 2.3.1. Level of Prior Knowledge and Training

This controlled experiment was differentiated primarily by subjects by collecting data from both beginners and skilled individuals. Beginners had no perception or experience of situational awareness, but had practiced with training scenarios before collecting data, and skilled individuals had operated the simulation platform at least five times. This experiment examines the extent to which mental models influence situational awareness in terms of the subjects’ basic conditions.

#### 2.3.2. Attention Resource Experiment

Attentional resource allocation is mainly governed by three aspects: information salience, information importance and information occurrence probability, and in this study, the effects of salience and information occurrence probability on attention and situational awareness are mainly investigated. The information in information salience is mainly divided into two parts, namely, visual information and auditory information. Visual information has different colors, sizes and codes, and its effects on attention are similar. Auditory information mainly includes the sound type and the size and duration of the warning tone. In the setting of the frequency of information appearance, this study analyzes the probability of information anomalies at different frequencies for comparison. The interface is divided into four regions to observe the eye movement behavior to judge the allocation of attention, as shown in Figure 3.

In the attention control experiment, it is mainly measured by monitoring and processing different parts of the interface, and three pieces of information are selected as the monitoring objects. The instrument information mainly includes the vehicle speed, oil pressure and oil volume. A vehicle speed of 20~30 km/h is within the normal range, an oil pressure of 200~500 kpa is within the normal range, and an oil volume of 20~50 L is within the normal range; if the instrument exceeds the normal value, the operator needs to press the corresponding key to eliminate it. The probability of information abnormalities is set as 1 min/time, 5 min/time and no abnormalities in the whole process, and the interval between the each abnormality setting is 4 s. The abnormal prompts are in red font by default. The information salience mainly depends on the font for comparison when the abnormality is separated into color changes and font size changes, as shown in Figure 4. The experiment converts the important affiliation degree of each information into an easy-to-perceive score according to a certain weight, which is convenient for the calculation of the attention model.

#### 2.3.3. Expectations

The simulated driving scenarios were divided into three kinds, in which the environment of plains and woods differed greatly, as did the frequency of the appearance of tanks and concealment locations, as shown in Figure 5. This experiment informed the subjects about environment-related information in advance and explored the level of situational awareness of the operators with certain psychological expectations.

#### 2.3.4. Degree of System Automation

In order to explore the influence of intelligent design and the information processing channel’s complexity on the situational awareness of armored vehicle crews, a comprehensive experiment on the factors influencing crew situational awareness was carried out based on the armored vehicle simulation platform using a two-factor independent comparison design. Factor 1 is the degree of intelligence of the system, including two levels of high and low intelligence, and different levels of intelligence are regulated by the degree of automation of the system that performs the same target task. Compared with the low intelligence level, the number of operation steps to accomplish the same target task is lower in the high intelligence level; i.e., some manual operation tasks are transferred to automatic system steps, such as automatic ammunition selection and loading [3].

Factor 2 is the information processing channel complexity, including three levels of visual single-channel processing, auditory single-channel processing and audio–visual dual-channel processing. Visual single-channel processing refers to the fact that when enemy obstacles appear, only the map in the lower right corner provides hints of type and orientation. Auditory single-channel processing refers to the interface without a map, relying only on sound to know that an obstacle has appeared. Dual-channel processing combines the above two cues.

### 2.4. Test Procedure

Before the experiment, the subjects put electrode pads on their upper body at designated locations, connected the cardiac and dermatoelectric devices, and checked whether the equipment, such as the eye-tracker, was working properly and debugged, and the participants were then ready to go. The experiment started with simulated driving, and it was randomly frozen in the middle so that the participants could fill out the SAGAT questionnaire, which was filled out from when the experiment continued to the end. Participants exported data and rested for 10 min to carry out the next group of experiments; the experimental process is shown in Figure 6.

### 2.5. Data Acquisition

Data collection was divided into two categories: one for subjective assessments and the other for objective performance and physiological signaling measures.

#### 2.5.1. Subjective Assessment

SART Questionnaire Score: The SART questionnaire evaluates the experimental feeling from four aspects: attentional resource demand and supply, comprehensive understanding of the experiment, and the driver’s state and state of mind. The armored vehicle occupants answered the questionnaire questions on the above four aspects after finishing the simulation experiment. The SART questionnaire score = comprehensive understanding of the scenario-(demand for attentional resources-supply of attentional resources)-other influencing factors.

SAGAT Questionnaire Correct: SAGAT Questionnaire Correct Response Score: The SAGAT is a memory-exploration-based measure used to comprehensively assess an operator’s level of situational awareness. It measures memory content by asking subjects to recall and report relevant information and is suitable for freezing tasks in simulation experiments. The correct response rate reflects the level of situational awareness. The operator drives a special vehicle in a simulated scenario and performs a task. At some point, the simulation is paused and the situational awareness questionnaire is displayed, which asks questions about the operator’s knowledge of the information in the scenario and records the reaction time, after which the simulation is resumed and the operator continues to drive. Finally, the operator’s answers were compared with the actual scenario information to score the results.

#### 2.5.2. Objective Performance

Annihilation time: the time between the appearance of the obstacle and its elimination by the operator; a shorter time proves that the level of situational awareness of the subjects is higher.

Operation correctness: After a complete set of combat missions, the system will automatically give the correctness rate of actions, including selection and loading, to measure the operator’s mission performance level.

Hit condition: this metric gives the accuracy of the shot, providing the exact hit rate.

#### 2.5.3. Eye movements and Physiological Signals

Blink frequency (Hz): The blink frequency can quantify the operator’s attention to the time period; the more the operator blinks in a certain time zone, the higher their attention to the scene at this time. The more environmental information is acquired, the fewer the blinks and the lower the attention to this time, and less environmental information is acquired.

Areas of interest (AOIs): by analyzing the areas of interest, it is possible to know in which area of the system interface the information affects the subject’s level of operational and situational awareness more.

Pupil diameter: The pupil diameter is one of the most commonly used indicators in eye tracking. The size of the pupil diameter can be used to determine the operator’s brain load state and attention level at that time. The larger the pupil diameter, the more focused the subject’s attention at that moment, and vice versa.

Eyelid opening: eyelid opening can be used to determine the operator’s level of situational awareness by knowing which time period in which they are more attentive.

## 3. Results

Statistical analysis of the various indicators was carried out using data statistical methods, using the 95% confidence level, to calculate the significance under the same variable and the correlation between different measurements and the results of the model calculations. The results of the analysis were analyzed graphically and conclusions were drawn.

### 3.1. Level of Prior Knowledge and Training


Experience is recognized as a key factor influencing operator situational awareness. Through learning and training, operators develop an understanding of the mental modeling system and the task. The maturation of the mental model leads to the development of “automatic” mechanisms. The present study explored the influence of this variable by distinguishing between the experience levels of the subjects.Table 1 shows the scores on the SART and SAGAT questionnaires and the performance of the beginners and the skilled participants. It is evident in Figure 7 that the skilled individuals significantly outperformed the beginners in terms of both SART questionnaire and SAGAT questionnaire scores and operational performance. This difference reflects the ability of skilled individuals to make more effective decisions by utilizing memorized mental models and experiences to react and process perceived information quickly.


Figure 8, below, shows the eye movement metrics of beginners and skilled individuals; beginners have larger eyelid openings and pupil diameters than skilled individuals but smaller blink frequencies than skilled individuals. No visual information can be acquired during eye hopping, so too high blink frequency implies an increased probability of losing the target, which suggests that the skilled are more efficient in acquiring cognitive resources [21]. The empirical comparison - eye tracking data are shown in Table 2.

For one category of comparison experiments, ANOVA and T-tests were performed on all the above measurements, leading to a significance analysis, as shown in Table 3. The significance test results indicated that the data for the experience level comparison were not statistically significant, which may imply that there are factors other than experience that influence the situational awareness of the operators.

### 3.2. Attention Resources

In this experiment, we first set the probability of abnormal information as once every minute, once every 5 min and no abnormality, as shown in Table 4, which shows the SART questionnaire and SAGAT questionnaire scores and operational performance. As in Figure 9 (left), the horizontal axis is the frequency of abnormal information and the vertical axis is the SAGAT and SART scores, respectively.

The eye movement data measured by the eye tracking device are shown in Table 5, with the horizontal coordinate being the anomaly frequency, the left axis being the blink frequency, the center axis being the eyelid opening, and the right axis being the pupil diameter. It can be seen that the frequency of anomalous information is 5 min/time and the eyelid opening is located in a trough, while the blink frequency is located in the peak and trough values. This represents that it is easier for the operator to acquire cognitive information at this time and the level of situational awareness is high. In contrast, at an abnormal frequency of 1 min/trial, as the visual information processing load increased, the subjects endeavored to observe the information components to better perform the task. As shown by an increase in eyelid opening and pupil diameters and a decrease in the number of blinks, the subject needed to recognize and comprehend the increased information, and the subject’s brain load level increased, making it difficult to maintain a high level of situational awareness. The probability of information anomaly - eye movement line graph are shown in Figure 10.

In addition, data on the viewed area of interest were collected and are shown in Table 6, which was plotted as a hotspot map. As can be seen in Figure 11, when the frequency of information anomalies decreases, the percentage of gazes directed at the instrument decreases significantly, while the percentage of gazes directed at the aiming center increases, indicating that the operator is able to focus more on accomplishing the combat task. Information anomaly probability—Significance test is shown in Table 6. The information anomaly probability—significance test data are shown in Table 7.

In the setting of information salience, the main focus is on visual information, and when the information is abnormal, it is categorized into reddening of the font and making the font larger. As seen in Figure 12, the questionnaire score is significantly lower when the font becomes larger. The information salience—questionnaire score and performance data are shown in Table 8.

The eye movement data showed that all indicators were elevated when the font size of the anomalous message became larger, implying that the message limited the cognitive ability of the operator and that more of the operator’s attention had to be focused on the instrumentation area while completing the typical task, increasing the operator’s mental load and making situational awareness lower, consistent with the questionnaire results. The information salience—eye tracking data are shown in Table 9. The information salience—eye movement line graph is shown in Figure 13.

### 3.3. Expectations

By informing the operator in advance about the possible situations in the scenario, the subjects’ indicators also changed. In the SART questionnaire and the SAGAT questionnaire, the level of performance failed to improve despite the operator’s pre-expectations, which were influenced by the difficulty of the task itself. The expectations questionnaire—score and performance data are shown in Table 10. The expectations questionnaire - score and performance bar chart are shown in Figure 14. The expectations—eye movement data are shown in Table 11. The expectation-eye movement bar chart is shown in Figure 15.

### 3.4. System Automation

The effects of different cueing channels on situational awareness were assessed by assessing the subjects’ SART and SAGAT scale scores, as well as operational performance, under different information processing channel conditions. The mean results are shown in Table 12, and the trends in these four data sets under different cueing channels are shown in Figure 16.

As can be seen in Table 12 above, the SART questionnaire and SAGAT questionnaire scores were significantly higher in the condition were enemy obstacles were only indicated by an auditory warning, whereas the SART questionnaire scores were significantly lower in the audio-visual dual-channel condition. In terms of typical performance, the operation and hit probabilities were significantly higher in the audio–visual dual-channel condition.

The results of the descriptive statistics of the eye movement measurement index are shown in Table 13. As seen in Figure 17, both visual and auditory alarms were present when an enemy obstacle appeared, when subjects had smaller eyelid openings and pupil diameters, i.e., when it was easier to access cognitive resources. A higher blinking frequency indicated that more information about the environment was accessed. Compared to the audio–visual dual channel, the single channel had a significantly higher pupil diameter and eyelid opening, representing an increased level of subjective brain load, a lower blink frequency, a higher difficulty of accessing resources, and a greater subjective load. The cue channel-eye movement line graph is shown in Figure 17.

This study further analyzed the distribution of the operator’s gaze area of interest under different cueing conditions. A small map area, aiming area, and instrument area were set as areas of interest to assess the effects of visual and auditory cues on the distribution of attention. It can be seen that under the visual cue condition, the percentage of gazes focused on the mini-map region is the largest; when only auditory cues are available, there is no longer a mini-map, human eye movement behaviors converge more on the instrument region, and attention on the aiming region of the main interface increases.

According to the hotspot in Figure 18, it can be seen that when there is only a visual cue, the focus of the gaze is in the center area and the mini-map area; when there is a dual audio–visual cue, the focus of the gazes is mostly the center area and the gaze is extended to the instrument area; and when there is an auditory cue, the focus of the gazes mainly converged only in the center area. Due to the above analysis of the eye movement results, when only auditory cues are present, the operator’s task becomes more difficult, the brain load becomes larger, and they may lack the residual energy to eliminate the instrumentation anomalies, which can explain the phenomenon of map hotspots. The prompt channel—interest area gaze proportion data are shown in Table 14. The cue channel—interest area gaze proportion bar chart is shown in Figure 18. The heat map—interest area gaze proportion is shown in Figure 19.

The physiological measurements were processed mainly for the electrocorticographic data, and the curves are shown in Figure 20. The audio–visual cueing condition had the weakest fluctuations in the electrodermal signals, whereas the visual cueing alternated between strong and weak electrodermal signals, and most of the electrodermal signals were at a high level in the auditory condition.

In the comparison of intelligence, this experiment is mainly differentiated regarding the selection and loading of bullets. The SART and SAGAT questionnaire scores and performances are shown in Table 15 below, from which the results obtained from high- and low-intelligence systems are not significant. The automation levels—questionnaire scores and performance data are shown in Table 15.

From the eye movement data in Table 16, the blinking frequency when using the high-intelligence system rises, and the eyelid opening and pupil diameter decrease, which shows that after the reduction in operation steps, the operator has more energy to acquire environmental information, but the cognitive load of the operator in the execution of the task decreases, probably due to the lack of the observation of the bullet selection process. This phenomenon may be related to the lack of necessary observation and decision-making steps in the automation process, resulting in a relatively relaxed mental state of the operator during task execution, with a decrease in eye-movement metrics and a level of attention that is instead inferior to that of the low-intelligence system. The automation levels—eye movement line graph is shown in Figure 21.

In the electrodermal signal curve, for the high-intelligence system, there are clear regular peaks and valleys, proving that after dispensing with the operator’s self-judgment of the type of enemy obstacle vehicle, the operator’s behavior becomes a completely regular sequence, which also means that the process of thinking and decision-making regarding this action is missing, and it evolves into a mechanized sequence of moving, aiming, and shooting actions. The Automation levels—Electrodermal Signal are shown in Figure 22.

## 4. Discussion

### 4.1. Influence of Experience

The results of the study showed that experienced operators showed higher levels of both subjective and objective measures of situational awareness; e.g., the SAGAT scores improved by 7.75 and the SART scores improved by 5.87. This finding supports the importance of experience for improving situational awareness. However, abnormalities in physiological data, particularly eye movement indices, suggest that even skilled individuals may experience a decrease in vigilance when performing tasks for extended periods of time. This suggests that even experienced operators need regular vigilance training and assessments in high-risk operating environments.

### 4.2. Effects of Expectations

The effect of expectation setting on situational awareness was not significant. Although there was an increase in eye movement metrics when the participants expected an increased number of barriers, this did not directly translate into increased levels of situational awareness. This may indicate that expectation setting alone is not sufficient to cope with dynamic changes in complex environments and that operators may need more real-time information and feedback to adapt to changes in the environment.

### 4.3. Effects of Automation

Automated, controlled experiments revealed opposing results. When the cueing channel was dual-channel, the level of situational awareness was higher; e.g., the correctness rate increased by 4.75% and 6.08% and the eyelid opening decreased by 1.744 mm and 2.081 mm, respectively, compared with the single-channel condition and the eye-movement area of interest of participants’ gazes was mainly focused on the main interface of the task. In contrast, in the single-channel condition with only auditory cues, it was more difficult for occupants to access cognitive information and they had lower levels of situational awareness. This suggests that multimodal information cues may help to improve operators’ situational awareness.

However, the high levels of automation may be due to the fact that automation reduces the number of steps that require autonomous decision-making by the occupant, making the operator’s task actions routine and thus reducing the level of situational awareness. For example, in the high-automation experiment, eyelid openings and pupil diameters decreased by 1.365 mm and 0.145 mm, respectively. This suggests that excessive automation may lead to the operator becoming relatively lax in performing tasks. This finding has important implications for the design of automated systems, suggesting that we need to reduce the operational burden while maintaining operator engagement and cognitive challenges.

## 5. Conclusions

The purpose of this study was to examine the role of situational awareness in armored vehicle operations and its relationship with operator experience, expectation setting, the information cueing style, and the system automation level. Through a series of experiments, we collected questionnaire scores, measured operational performance, and collected eye-tracking data to objectively assess multiple dimensions of situational awareness. Experienced operators had significantly higher levels of situational awareness, with SAGAT scores of 71.5 and SART scores of 17.43 for beginners, and SAGAT scores of 79.25 and SART scores of 23.3 for skilled operators. Operators with expectations instead had a lower level of situational awareness, with an SAGAT score of 79.92 and 98.5% when they expected a sparse spread of obstacles, and an SAGAT score of 71.58 and 98.5% when they expected a dense spread of obstacles. The level of situational awareness was negatively correlated with the information importance affiliation of the non-primary task and the frequency of abnormal information, with an SAGAT score of 70.5 and an eyelid opening of 9.491 mm when the probability of information abnormality was 1 min/trip and an SAGAT score of 81.25 and an eyelid opening of 7.032 mm when the probability of information abnormality was 5 min/trip. The level of situational awareness was higher for dual-channel cues than for single-channel cues, with a blink frequency of 60.251 Hz and an eyelid opening of 7.682 mm for visual–auditory dual cues, a blink frequency of 60.203 Hz and an eyelid opening of 9.426 mm for visual-only cues, and a blink frequency of 60.233 Hz and an eyelid opening of 9.763 mm for auditory-only cues. A high situational awareness is directly proportional to the level of intelligence, with an SAGAT score of 83.33 and an eyelid opening of 7.518 for the high-intelligence system, and an SAGAT score of 73.25 and an eyelid opening of 8.883 for the low-intelligence system.

The results of this study are critical to understanding the role of situational awareness in high-risk environments. Particularly in mission-critical areas such as armored vehicle operations, increasing the situational awareness of operators can significantly improve mission efficiency and safety. In addition, the results of this study provide valuable insights into how to design cues and anomaly alerts and how to balance automation and operator engagement when designing highly intelligent systems.

## Figures and Tables

**Figure 1 sensors-24-03688-f001:**
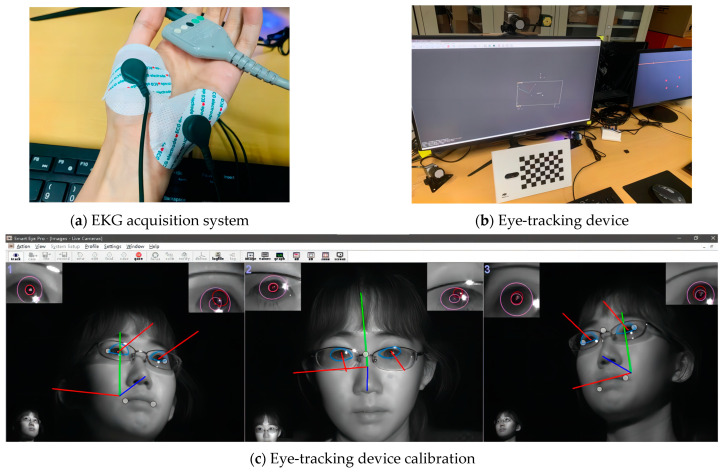
Test instruments and equipment.

**Figure 2 sensors-24-03688-f002:**
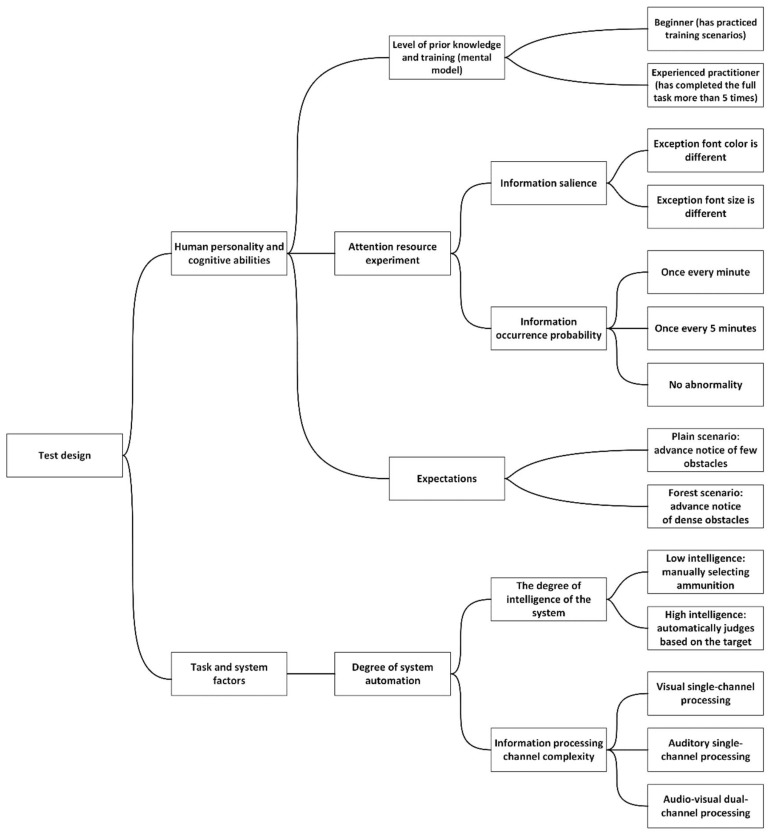
Test design.

**Figure 3 sensors-24-03688-f003:**
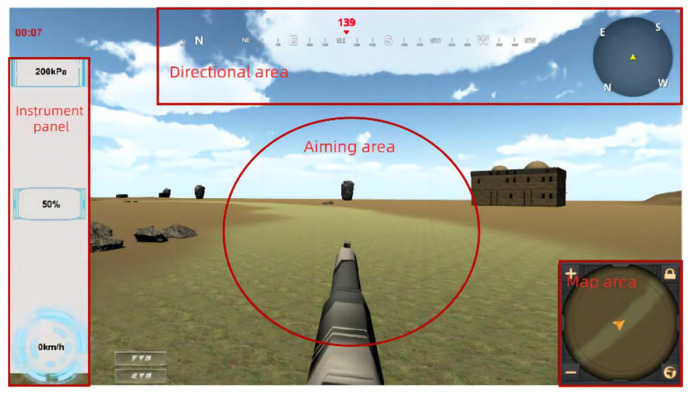
Interface area division.

**Figure 4 sensors-24-03688-f004:**
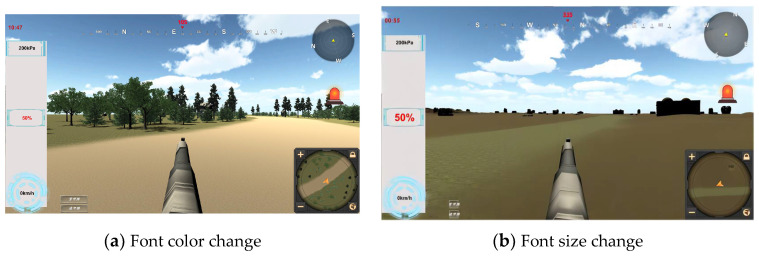
Different fonts.

**Figure 5 sensors-24-03688-f005:**
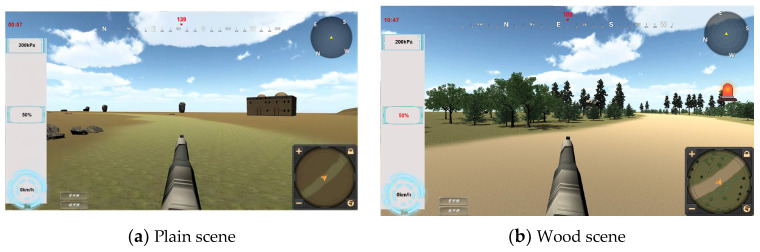
Expectations—different scenes.

**Figure 6 sensors-24-03688-f006:**
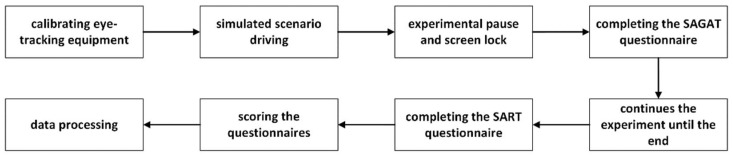
Experimental flowchart.

**Figure 7 sensors-24-03688-f007:**
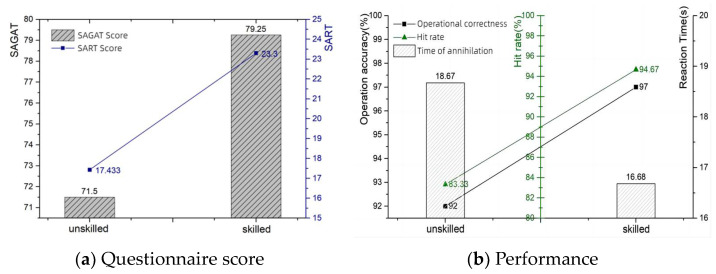
Experience comparison.

**Figure 8 sensors-24-03688-f008:**
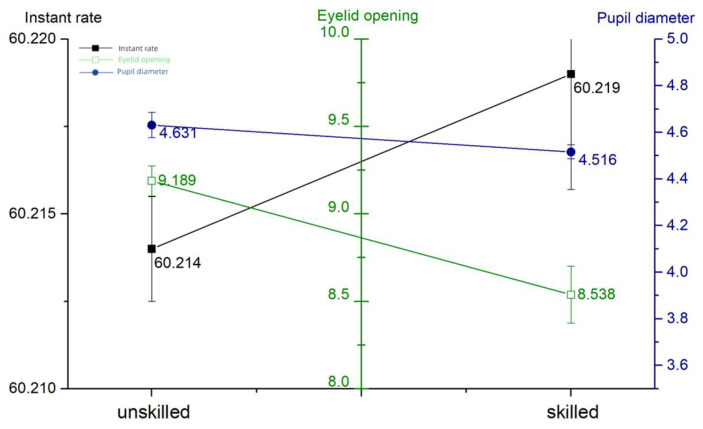
Experience comparison—eye movement.

**Figure 9 sensors-24-03688-f009:**
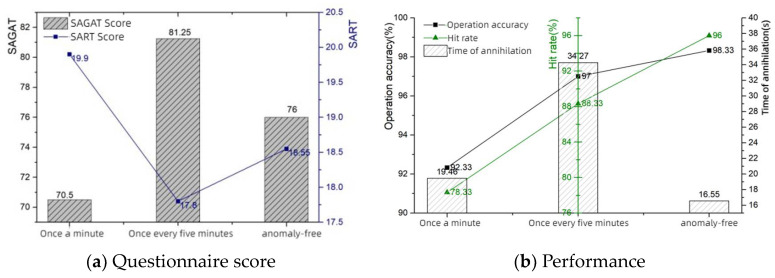
Information anomaly probability.

**Figure 10 sensors-24-03688-f010:**
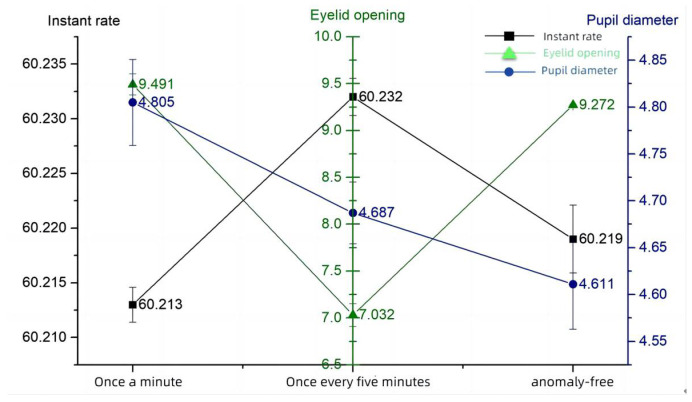
Information anomaly probability—eye movement.

**Figure 11 sensors-24-03688-f011:**
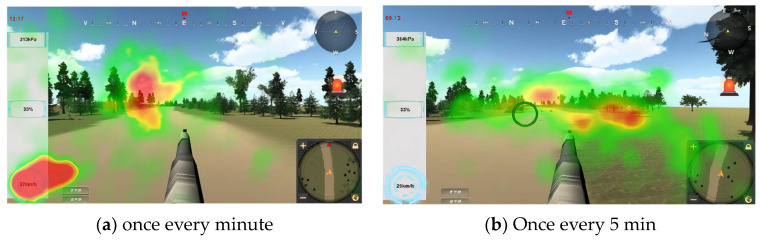
Heat map—different information anomaly probability.

**Figure 12 sensors-24-03688-f012:**
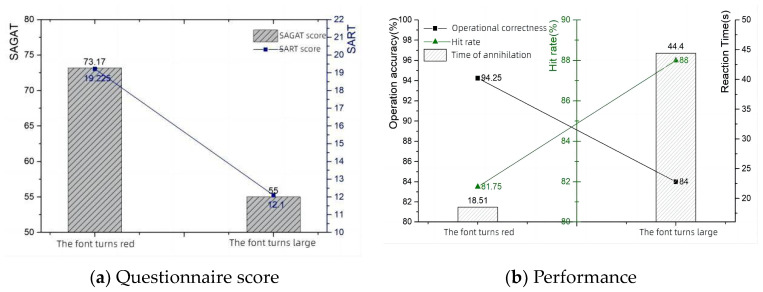
Information salience—questionnaire score and performance.

**Figure 13 sensors-24-03688-f013:**
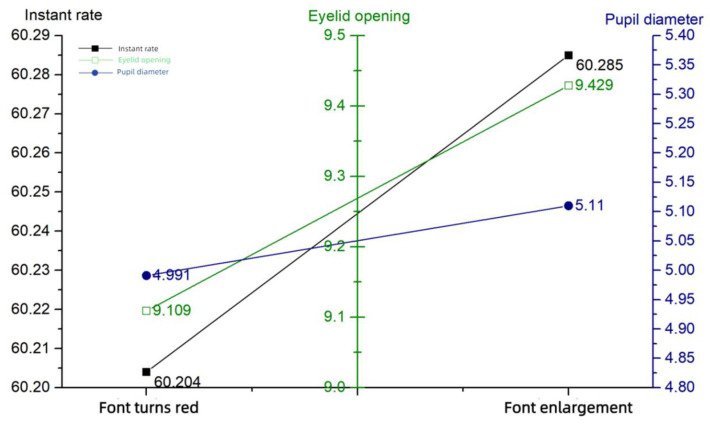
Information salience—eye movement.

**Figure 14 sensors-24-03688-f014:**
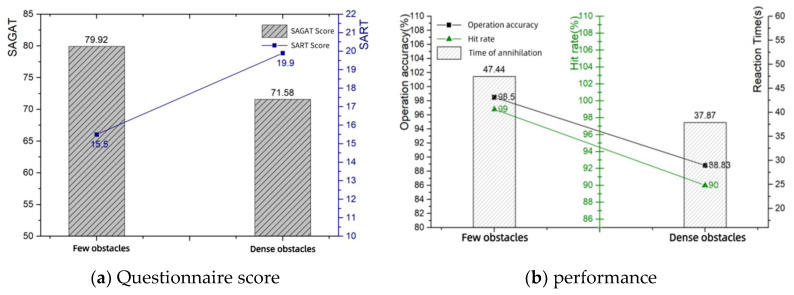
Expectations questionnaire—score and performance.

**Figure 15 sensors-24-03688-f015:**
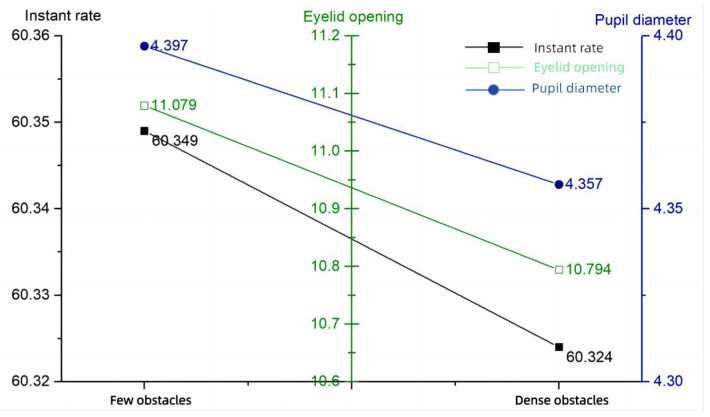
Expectations—eye movement.

**Figure 16 sensors-24-03688-f016:**
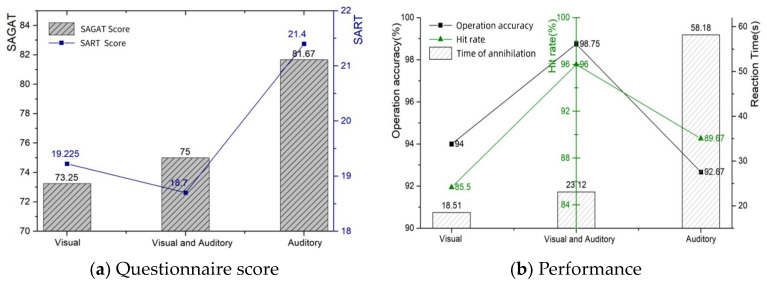
Prompt channel—questionnaire score and performance.

**Figure 17 sensors-24-03688-f017:**
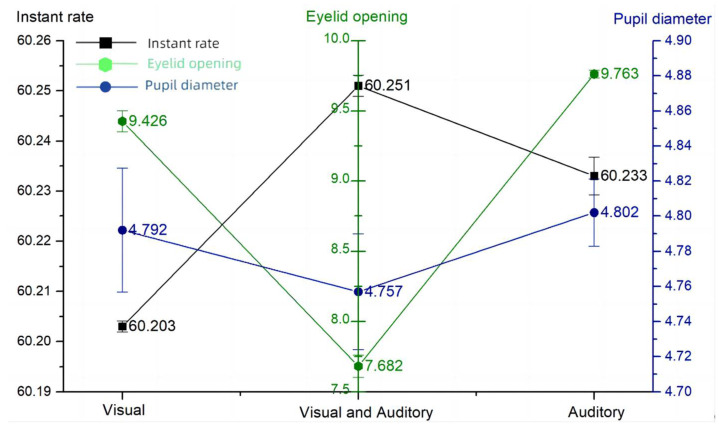
Cue channel—eye movement.

**Figure 18 sensors-24-03688-f018:**
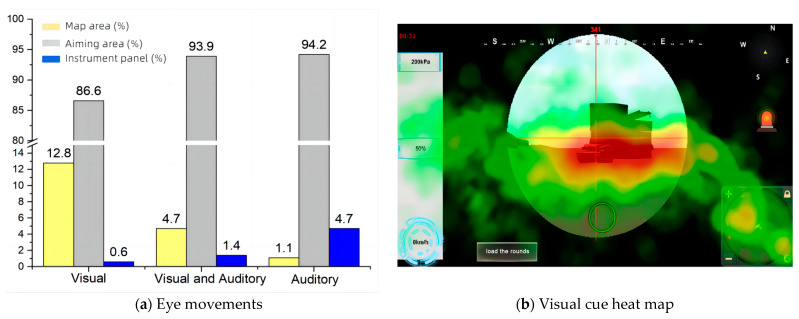
Cue channel—interest area gaze proportion.

**Figure 19 sensors-24-03688-f019:**
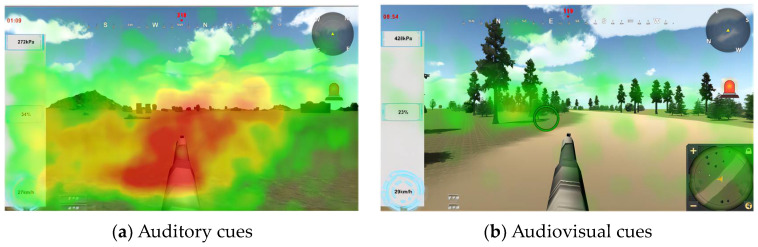
Heat map—interest area gaze proportion.

**Figure 20 sensors-24-03688-f020:**
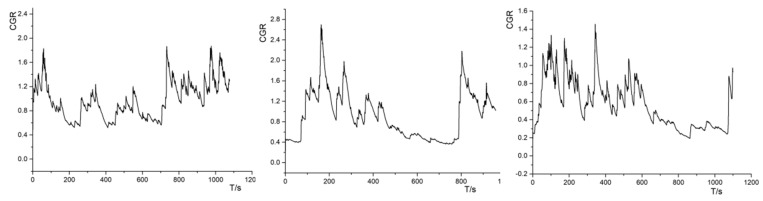
Cue channel electrodermal signal—visual cue (**left**), audiovisual cue (**middle**), and auditory cue (**right**).

**Figure 21 sensors-24-03688-f021:**
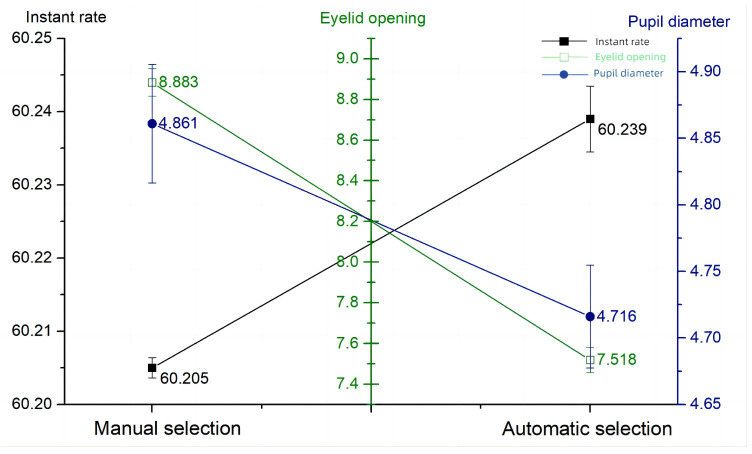
Automation levels—eye movement.

**Figure 22 sensors-24-03688-f022:**
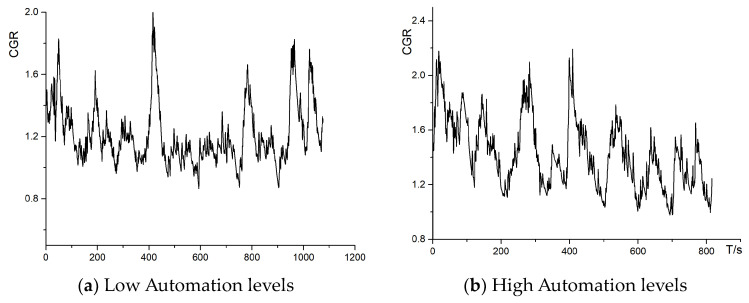
Automation levels—electrodermal signal for (**a**) low automation levels and (**b**) high automation levels.

**Table 1 sensors-24-03688-t001:** Experience comparison—questionnaire scores and performance.

	SAGAT Score	SART Score	Operational Correction	Hit Rate	Time of Annihilation
unskilled	71.5	17.43	92%	83.33%	18.67 s
skilled	79.25	23.3	97%	94.67%	16.68 s

**Table 2 sensors-24-03688-t002:** Empirical comparison—eye tracking data.

	Blink Frequency (Hz)	Eyelid Opening (mm)	Pupil Diameter (mm)
unskilled	60.214 ± 0.015	9.189 ± 0.855	4.631 ± 0.539
skilled	60.219 ± 0.033	8.538 ± 1.626	4.516 ± 0.301

**Table 3 sensors-24-03688-t003:** Empirical control—significance test.

	Analysis of Variance		T-Test	
	F	Significance	T	Sig. (Two-Tailed)
Operation accuracy	0.400	0.561	−1.936	0.125
Hit rate	2.298	0.204	−1.176	0.305
Time of annihilation	0.309	0.608	1.135	0.320
Blink frequency	2.802	0.169	−0.139	0.896
Eyelid opening	1.356	0.309	0.354	0.741
Pupil diameter	2.086	0.222	−0.581	0.593

**Table 4 sensors-24-03688-t004:** Information anomaly probability—questionnaire scores and performance.

Information Anomaly Probability	SAGAT Score	SART Score	Operation Accuracy	Hit Rate	Time of Annihilation
once every minute	70.5	19.9	92.33%	78.33%	19.46 s
once every 5 min	81.25	17.8	97%	88.33%	34.27 s
no abnormality	76	18.55	98.33%	96%	16.55 s

**Table 5 sensors-24-03688-t005:** Information anomaly probability—eye movement data.

Information Anomaly Probability	Blink Frequency (Hz)	Eyelid Opening (mm)	Pupil Diameter (mm)
once every minute	60.213 ± 0.016	9.491 ± 1.130	4.805 ± 0.460
once every 5 min	60.232 ± 0.017	7.032 ± 1.210	4.687 ± 0.332
no abnormality	60.219 ± 0.031	9.272 ± 0.789	4.611 ± 0.476

**Table 6 sensors-24-03688-t006:** Information anomaly probability—interest area gaze proportion.

Once Every Minute		Once Every 5 Min	
Area of Interest	Proportion of Gaze (%)	Area of Interest	Proportion of Gaze (%)
Map area	3.1	Map area	1.9
Aiming area	81.7	Aiming area	84.3
Instrument panel	11.7	Instrument panel	9.4
Directional area	3.5	Directional area	3.4

**Table 7 sensors-24-03688-t007:** Information anomaly probability—significance test.

	Analysis of Variance		T-Test	
	F	Significance	T	Sig. (Two-Tailed)
Operation accuracy	0.492	0.522	−1.673	0.170
Hit rate	0.752	0.775	−1.551	0.196
Time of annihilation	10.068	0.034	−1.549	0.196
Blink frequency	0.002	0.968	−0.834	0.451
Eyelid opening	0.094	0.775	1.485	0.212
Pupil diameter	0.613	0.477	0.208	0.846

**Table 8 sensors-24-03688-t008:** Information salience—questionnaire score and performance.

Abnormal Information Display	SAGAT Score	SART Score	Operation Accuracy	Hit Rate	Time of Annihilation
Font turns red	73.17	19.225	94.25%	81.75%	18.51
Font turns large	55	12.1	84%	88%	44.4

**Table 9 sensors-24-03688-t009:** Information salience—eye tracking data.

Information Anomaly Probability	Blink Frequency (Hz)	Eyelid Opening (mm)	Pupil Diameter (mm)
Font turns red	60.204	9.109	4.991
Font turns large	60.285	9.429	5.110

**Table 10 sensors-24-03688-t010:** Expectations questionnaire—score and performance.

Expectations	SAGAT Score	SART Score	Operation Accuracy	Hit Rate	Time of Annihilation
Few obstacles	79.92	15.5	98.5%	99%	47.44 s
Dense obstacles	71.58	19.9	88.83%	90%	37.87 s

**Table 11 sensors-24-03688-t011:** Expectations—eye movement.

Expectations	Blink Frequency (Hz)	Eyelid Opening (mm)	Pupil Diameter (mm)
Few obstacles	60.349	11.079	4.397
Dense obstacles	60.324	10.794	4.357

**Table 12 sensors-24-03688-t012:** Prompt channel—questionnaire score and performance.

	SAGAT Score	SART Score	Operation Accuracy	Hit Rate	Time of Annihilation
Visual Cues	73.25	19.225	94%	85.5%	18.51 s
Audiovisual Cues	75	18.7	98.75%	96%	23.12 s
Auditory Cues	81.67	21.4	92.67%	89.67%	58.18 s

**Table 13 sensors-24-03688-t013:** Prompt channel—eye movement.

	Blink Frequency (Hz)	Eyelid Opening (mm)	Pupil Diameter (mm)
Visual Cues	60.203 ± 0.011	9.426 ± 0.755	4.792 ± 0.352
Audiovisual Cues	60.251 ± 0.021	7.682 ± 0.784	4.757 ± 0.330
Auditory Cues	60.233 ± 0.038	9.763 ± 0.288	4.802 ± 0.190

**Table 14 sensors-24-03688-t014:** Prompt channel—interest area gaze proportion.

Visual Cues		Audiovisual Cues		Auditory Cues	
Area of Interest	Proportion of Gaze (%)	Area of Interest	Proportion of Gaze (%)	Area of Interest	Proportion of Gaze (%)
Map area	12.8	Map area	4.7	Map area	1.1
Aiming area	86.6	Aiming area	93.9	Aiming area	94.2
Instrument panel	0.6	Instrument panel	1.4	Instrument panel	4.7

**Table 15 sensors-24-03688-t015:** Automation levels—questionnaire scores and performance.

	SAGAT Score	SART Score	Operational Correctness	Hit Rate	Time of Annihilation
low automation levels	73.25	19.225	94.4%	85.4%	18.51 s
high automation levels	83.33	19.2	87.33%	87.33%	31.82 s

**Table 16 sensors-24-03688-t016:** Automation levels—eye movement data.

	Blink Frequency (Hz)	Eyelid Opening (mm)	Pupil Diameter (mm)
low automation levels	60.205 ± 0.014	8.883 ± 0.678	4.861 ± 0.445
high automation levels	60.239 ± 0.045	7.518 ± 0.613	4.716 ± 0.386

## Data Availability

The raw data supporting the conclusions of this article will be made available by the authors on request.

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
