# Peer review of "Factors Affecting the Situational Awareness of Armored Vehicle Occupants"

_sensors, 2024, doi:10.3390/s24113688_

Round 1

Reviewer 1 Report

Comments and Suggestions for Authors

This study provides a comprehensive investigation of the factors influencing situational awareness (SA) in armored vehicle operations. The authors employed a combination of experimental methods, including questionnaires, performance assessments, and eye-tracking, to gather data from a group of participants. The findings highlight the significant impact of experience, expectation setting, information cueing style, and system automation level on SA. The study offers valuable insights for improving training programs, interface design, and automation strategies to enhance SA and consequently, operational efficiency and safety in armored vehicle operations.

The study employs a multifaceted approach that combines questionnaires, performance assessments, and eye-tracking to provide a comprehensive evaluation of SA. The methodology of the study is clearly described. The results are presented in a clear and concise manner. The findings offer practical implications for improving training programs, interface design, and automation strategies to enhance SA in armored vehicle operations.

The authors should acknowledge and discuss any limitations of their study, such as potential sample bias or external validity concerns. Suggest potential directions for future research to further explore the complexities of SA in armored vehicle operations and other high-risk environments.

Additional comments:

·        Figure 8a does not have a legend.

·        Figure 13b is missing.

·        Figure 13a does not have an axis label.

·        The references do not conform to the journal's requirements.

Author Response

We have made changes based on the comments you made, we have added a legend to figure 8 and axis labels to figure 13. Also Fig. 13 has only one picture without Fig. 13b, so there is no lack of Fig. 13b, added the limitations of the research in this paper as well as directions for future research, and modified the references according to the requirements of the journal

Reviewer 2 Report

Comments and Suggestions for Authors

The article covers extensive research on factors influencing situational awareness in armored vehicle users. The topic is interesting but presented in a chaotic manner. The first question that arises pertains to the research group, which consists of professionals aged 20 to 30. The mean age and standard deviation were not provided. Can we truly consider individuals in this age range to have significant professional experience? The conclusions formulated do not deviate from those obtained in studies regarding automation, or information modalities, for example. A new aspect is the research object. Unfortunately, the article contains numerous, to put it mildly, editorial oversights. The authors failed to properly reference tables and figures, and also confused the numbering. One figure is even missing, despite having a caption. A few examples: line 179, 190 - the same numbers and titles; line 253 and 262 - fig. 7; line 277 should be Fig. 9 but is Fig. 8; missing figure 12; line 398 - Table 17 and verse 408 Table 8; referencing other tables and figures in the text; inconsistent line spacing; missing figure 13b. These are just examples, and there are more instances. The reference to the ethics committee document allowing research involving human participants is missing.

In its current form, the article is not suitable for publication.

Author Response

In response to these questions you raised, we have made the following changes:
(1) Provided the mean age and standard deviation of study group members.
(2) Removed heading 2.4 from line 190.
(3) Added figure legends for figure 8, figure 10, figure 13, figure 15, figure 17, and figure 18(a).
(4) Changed line spacing to be consistent throughout the text.
(5) Figure 13 has only one image, not Figure 13b
(6) Modified the formatting of the references according to the format of the MDPI paper.
(7) Added an analysis of the experimental results data in the Discussion and Conclusion section and corresponded to the previous experimental conclusions.

Round 2

Reviewer 2 Report

Comments and Suggestions for Authors

Thank you for considering the comments. I think the quality of the article has benefited from it.